# MULTILEVEL XAI: VISUAL AND LINGUISTIC BONDED EXPLANATIONS

## ABSTRACT

Applications of deep neural networks are booming in more and more fields but lack transparency due to their black-box nature. Explainable Artificial Intelligence (XAI) is therefore of paramount importance, where strategies are proposed to understand how these black-box models function. The research so far mainly focuses on producing, for example, class-wise saliency maps, highlighting parts of a given image that affect the prediction the most. However, this way does not fully represent the way humans explain their reasoning and, awkwardly, validating these maps is quite complex and generally requires subjective interpretation. In this article, we conduct XAI differently by proposing a new XAI methodology in a multilevel (i.e., visual and linguistic) manner. By leveraging the interplay between the learned representations, i.e., image features and linguistic attributes, the proposed approach can provide salient attributes and attribute-wise saliency maps, which are far more intuitive than the class-wise maps, without requiring per-image ground-truth human explanations. It introduces self-interpretable attributes to overcome the current limitations in XAI and bring the XAI towards human-like level. The proposed architecture is simple in use and can reach surprisingly good performance in both prediction and explainability for deep neural networks thanks to the low-cost per-class attributes[1].

## 1 INTRODUCTION

Exciting developments in computational resources with a significant rise in data size have led deep neural networks (DNNs) to be widely used in various tasks, for example image classification. Despite their excellent performance in prediction, DNNs are seen as black boxes as their decision process generally includes a huge number of parameters and nonlinearities (Gilpin et al., 2018; Hagras, 2018; Zeiler & Fergus, 2014). The lack of explanation in these black boxes hinders their direct implementation in important and sensitive domains such as medicine and autonomous driving, where human life may directly be affected (Loyola-Gonzalez, 2019; Lipton, 2018).

An example would be the DNNs trained to detect coronavirus. Although many works have been conducted and claimed to have a high predictive performance in detecting COVID-19 cases, a Turing Institute's recent report (Heaven, 2021) disappointingly finds that Artificial Intelligence (AI) used to detect coronavirus had little to no benefit and may even be harmful, mainly due to unnoticed biases in the data and its inherent black-box nature (also see e.g. Roberts et al. 2021). Another example is a woman who was hit and killed by an autonomous car. An investigation showed that the death was caused by the incapability of the car in detecting a human unless they are near a crosswalk (McCausland, 2019). In addition to these life-related examples, there are plenty of others where bias in training data or the model itself causes unwanted discriminations that may immensely affect people's lives. Amazon's AI-enabled recruitment tool is an example of how discriminative these models could be by only recommending men and directly eliminating resumes including the word "woman"; the company later announced that this tool had never been used to recruit people due to the detected bias (Olavsrud, 2022). These examples clearly show that for machine learning models to gain acceptance, it is critical to be able to reason why a certain decision has been made to prevent any unwanted consequences.

---

[1]Our code webpage: `https://anonymous.4open.science/r/Multilevel_XAI-FBBC`

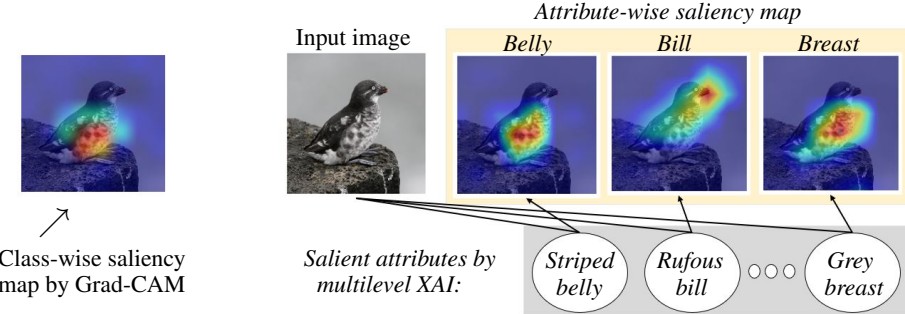

Figure 1: Explainability of the proposed multilevel XAI model. A bird image (*middle*) from Least Auklet class is predicted correctly by our approach, with human-like multilevel explanations via salient attributes (e.g. "striped belly") and the corresponding attribute-wise saliency maps (*right*). Result by Grad-CAM (Selvaraju et al., 2017) (*left*) is also given for comparison.

Explanations delivered by explainable AI (XAI) can help machine learning practitioners debug their models by for example investigating the misclassification cases (Adadi & Berrada, 2018) and detecting bias in data (Tan et al., 2017). There have been a number of works in this context to reveal the reasoning of the black-box models (Simonyan et al., 2014; Springenberg et al., 2014; Zhou et al., 2016; Chattopadhay et al., 2018; Petsiuk et al., 2018; Ribeiro et al., 2016; 2018). However, the most widely used techniques, creating class-wise saliency maps (e.g. see left of Figure 1) to indicate the areas that contribute to the prediction the most, have severe innate limitations. The first is the validation process of these maps, which is mostly qualitative or requires labour intensive object-wise annotations (Goebel et al., 2018; Park et al., 2018). A recent study in (Bearman et al., 2016) showed that a full supervision of object segmentation by humans takes around 78 seconds per instance while higher error rate bounding boxes take 10 seconds per instance to produce, which are much more expensive than 1 second per instance image level annotations. Moreover, requiring a higher level of annotation by experts is rather impractical. Another limitation stems from the discrepancy between these maps and human-like explanations. Humans naturally explain their reasoning using discriminative words (e.g. domestic *vs* wild or weak *vs* strong to differentiate a cat from a lion) together with pointing to where those words lie in the given image if visually permitting (Park et al., 2018; Goebel et al., 2018) (*cf.* our results on the right of Figure 1). To produce human-like explanations, this multilevel (i.e., visual and linguistic) manner is crucial, which also inspires the work in this article.

In this article we propose a new methodology called *multilevel XAI* to delve into DNNs by leveraging visual and linguistic attributes. Our approach exploits per-class attributes (rather than per-image attributes, which are too expensive and generally impractical) to interpret DNNs in e.g. classifying raw images. By creating multilevel explanations, i.e., linguistic salient attributes and attribute-wise saliency maps, our method can achieve towards human-like explanations (e.g. see right of Figure 1). This is a big step forward in XAI and this new methodology does not suffer from the above-mentioned limitations existing in current XAI solutions. The proposed setting adds a tiny extra cost to the training set, i.e., per-class attributes, which can be easily obtained if needed using for example online search engines or some autonomous tools (e.g. GPT-3 API Brown et al., 2020), and once acquired they can always be in use since in most cases they are time and image invariant.

Our main contributions lie in: i) proposing a multilevel XAI methodology which is easy to use and can achieve towards human-like explanations; ii) implementing extensive experiments on both coarse-grained and fine-grained datasets to validate the performance of the proposed approach; and iii) conducting insightful discussions in XAI and future paths.

## 2 METHODOLOGY OF MULTILEVEL XAI

In this section, we introduce our multilevel XAI methodology, see Figure 2 for its main architecture. It consists of three main components: i) a pre-trained feature extraction block generating high level image features from input images (left of Figure 2); ii) a self-explainable DNN block bridging the extracted features with linguistic attributes (middle of Figure 2); and iii) a language model block

(being frozen after training) linking the linguistic attributes to the output class labels (right of Figure 2). All of these blocks are important and are well studied in various fields, yet in the XAI regime, their study is rather limited. To the best of our knowledge, this is the first time they have been used to explain neural networks in a multilevel (i.e., visual and linguistic) manner particularly when the per-image attributes are unavailable. Further description is given below.

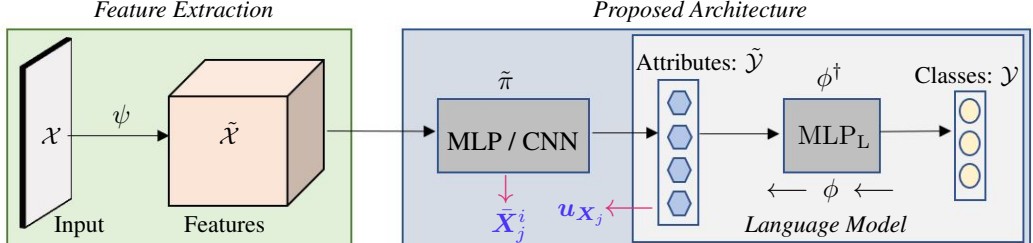

Figure 2: Multilevel XAI architecture. Image features $\tilde{\mathcal{X}}$ are extracted from images $\mathcal{X}$ using feature extraction model $\psi$. Class labels $\mathcal{Y}$ are embedded into class attributes $\tilde{\mathcal{Y}}$ using language model $\phi$. DNN $\tilde{\pi}$ (e.g. MLP and CNN) is then trained to match $\tilde{\mathcal{X}}$ with $\tilde{\mathcal{Y}}$. The explainability of DNN $\tilde{\pi}$ is by the obtained salient attributes $\boldsymbol{u}_{\boldsymbol{X}_j}$ (linguistic) and attribute-wise saliency maps $\bar{\boldsymbol{X}}_j^i$ (visual).

**Preliminary.** Let $\mathcal{X}$ be the set of images and $\mathcal{Y} = \{1, 2, \cdots, C\}$ be the set of $C$ class labels. Let $\mathcal{S} = \{(\boldsymbol{X}_i, y_i) \mid \boldsymbol{X}_i \in \mathcal{X}, y_i \in \mathcal{Y}, i = 1, 2, \ldots, N\}$ be a training set with $N$ image/label pairs, where $y_i$ is the ground truth label of image $\boldsymbol{X}_i \in \mathbb{R}^{M_1 \times M_2 \times M_3}$ with $M_3$ set to 1 and 3 respectively for grey and colour images. Image classification is to learn a function say $\pi : \mathcal{X} \to \mathcal{Y}$ under a loss function $\rho : \mathcal{Y} \times \mathcal{Y} \to \mathbb{R}$ such that the energy $\sum_{\boldsymbol{X}_i \in \mathcal{X}; y_i \in \mathcal{Y}} \rho(y_i, \pi(\boldsymbol{X}_i))$ is minimised.

Given a deep neural architecture, the main aim is to find proper weights, say $\boldsymbol{W}$, for all layers, such that $\pi(\boldsymbol{X}; \boldsymbol{W})$ is able to predict the label of $\forall \boldsymbol{X} \in \mathcal{X}$ accurately through training. If the quality of $\mathcal{S}$ is perfect, it is highly likely that $\pi(\boldsymbol{X}; \boldsymbol{W})$ can perform excellently after training by minimising

$$\sum_{(\boldsymbol{X}_i, y_i) \in \mathcal{S}} \rho(y_i, \pi(\boldsymbol{X}_i, \boldsymbol{W})). \tag{1}$$

However, it is generally impossible to acquire an ideal set $\mathcal{S}$, plus the reason why the trained $\pi(\boldsymbol{X}; \boldsymbol{W})$ works lacks interpretability, i.e., its so-called black-box nature. This motivated many XAI methods (e.g. Grad-CAM Selvaraju et al. 2017 with its result in Figure 1) to peek inside this kind of black box.

To improve the performance of $\pi$ in its prediction, more complicated models are proposed, e.g. the way of exploiting image- and label-embeddings. Let $\tilde{\mathcal{X}}$ and $\tilde{\mathcal{Y}}$ respectively be the sets of image features and linguistic class attributes (which can be generalised to other sources of side information such as taxonomy of classes (Tsochantaridis et al., 2005)). The image- and label-embeddings are to find proper embedding functions $\psi : \mathcal{X} \to \tilde{\mathcal{X}}$ and $\phi : \mathcal{Y} \to \tilde{\mathcal{Y}}$, respectively. Then the image classification task can be generalised to learn a function $\tilde{\pi} : \tilde{\mathcal{X}} \to \tilde{\mathcal{Y}}$ such that the energy $\sum_{\boldsymbol{X}_i \in \mathcal{X}} \rho(\phi(y_i), \tilde{\pi}(\psi(\boldsymbol{X}_i)))$ is minimised together with $\psi$ and $\phi$. Note that function $\tilde{\pi}$ is general. An appropriate form of it may be found by exploiting compatibility functions used by the zero-shot learning community (Akata et al., 2015); see e.g. Frome et al. (2013); Reed et al. (2016); Romera-Paredes & Torr (2015) with their own pros and cons.

In this work we are interested in studying a neural architecture for $\tilde{\pi}$ (i.e., DNN $\tilde{\pi}$) with its weights, $\tilde{\boldsymbol{W}}$, via minimising the energy

$$\sum_{(\boldsymbol{X}_i, y_i) \in \mathcal{S}} \rho(\phi(y_i), \tilde{\pi}(\psi(\boldsymbol{X}_i), \tilde{\boldsymbol{W}})). \tag{2}$$

The choice of $\psi$ and $\phi$ is wide and can also be represented using neural architectures in deep learning. Our proposed architecture in Figure 2 is to reveal the explainability for $\tilde{\pi}$ utilising language models (for $\phi$) in a multilevel manner.

**Feature extraction.** Within our framework there is a freedom to choose the feature extraction models $\psi$ (see the left of Figure 2). To illustrate this flexibility we have used both pre-trained ResNet101 (He et al., 2016) and VGG16 (Simonyan & Zisserman, 2014) for visual feature extraction.

## 2.1 CLASS EMBEDDING

A central component in our approach is the introduction of a meaningful $K$-dimensional embedding space, $\tilde{\mathcal{Y}}$. We consider a mapping $\phi$ from class $y_i \in \mathcal{Y}$ to an embedding vector $\tilde{\boldsymbol{y}}_i \in \tilde{\mathcal{Y}}$, where each component of $\tilde{\boldsymbol{y}}_i$ is a linguistic attribute. In practice $\phi$ would be a probabilistic embedding describing the conditional probability of $\mathbb{P}(\tilde{\boldsymbol{y}}_i \mid y_i)$; however, obtaining this is difficult. Instead, we start from a matrix $\mathbf{A} \in \mathbb{R}^{C \times K}$ provided by experts (in our case this was conveniently provided by the zero-shot learning community, see Table 1 for an example Lampert et al. 2013) which can be interpreted as $\mathbb{E}(\tilde{\boldsymbol{y}}_i \mid y_i)$. In our approach we need the "inverse mapping", $\phi^\dagger(\tilde{\boldsymbol{y}}_i)$, giving $\mathbb{P}(y_i \mid \tilde{\boldsymbol{y}}_i)$. We learn this mapping using a multilayer perceptron (MLP; $\mathrm{MLP_L}$ in our model, see the right of Figure 2), where our inputs are noisy vectors $\tilde{\boldsymbol{y}}_i$ (i.e., the rows of matrix $\mathbf{A}$) and our targets are the classes $y_i$. This mapping is learned entirely without seeing the training images and is then frozen. Although this is a rather simple approach, it is extremely fast to learn and leads to good performance.

## 2.2 EXPLAINABLE NEURAL NETWORKS

Below we introduce the strategies regarding how the DNN $\tilde{\pi}$ in our proposed multilevel architecture (middle of Figure 2) can be explainable. Note that $\tilde{\pi} : \psi(\boldsymbol{X}_i) \to \phi(y_i)$, where $\boldsymbol{X}_i \in \mathcal{X}$ and $y_i \in \mathcal{Y}$. Since $\phi(y_i)$ is not unique, we train $\tilde{\pi}$ to learn the match between features and attributes in an unsupervised way (regarding $\tilde{\mathcal{Y}}$) using the training set $\mathcal{S}$ (i.e., image/label pairs), with the trained $\mathrm{MLP_L}$ and pre-trained $\psi$.

$\forall \boldsymbol{X}_j \in \mathcal{X}$ used for test, let $y_{\boldsymbol{X}_j} = \phi^\dagger(\tilde{\pi}(\psi(\boldsymbol{X}_j)))$, i.e., the predicted class label of the test image $\boldsymbol{X}_j$. Let $\tilde{\pi}(\psi(\boldsymbol{X}_j))_k$ be the $k$-th attribute of $\tilde{\pi}(\psi(\boldsymbol{X}_j))$, where $k \in \{1, 2, \ldots, K\}$. We define $\boldsymbol{u}_{\boldsymbol{X}_j} \in \mathbb{R}^K$ as the importance of the attributes for the test image $\boldsymbol{X}_j$ and evaluate it by taking the gradient of the predicted class label $y_{\boldsymbol{X}_j}$ with respect to every attribute of $\tilde{\pi}(\psi(\boldsymbol{X}_j))$, i.e.,

$$\boldsymbol{u}_{\boldsymbol{X}_j} = (u_{\boldsymbol{X}_j}^1, u_{\boldsymbol{X}_j}^2, \cdots, u_{\boldsymbol{X}_j}^K) = \left( \frac{\partial y_{\boldsymbol{X}_j}}{\partial \tilde{\pi}(\psi(\boldsymbol{X}_j))_1}, \frac{\partial y_{\boldsymbol{X}_j}}{\partial \tilde{\pi}(\psi(\boldsymbol{X}_j))_2}, \cdots, \frac{\partial y_{\boldsymbol{X}_j}}{\partial \tilde{\pi}(\psi(\boldsymbol{X}_j))_K} \right). \quad (3)$$

Then the top $K^*$ largest of $\{u_{\boldsymbol{X}_j}^k\}_{k=1}^K$ will be selected as the *salient linguistic attributes*. In this sense, $\tilde{\pi}$ therefore can be explained by these salient linguistic terms. As examples, two of the most common neural networks – MLP and CNN – are adopted for $\tilde{\pi}$ below.

**Explainable MLP.** When $\tilde{\pi}$ represents an MLP, say $\tilde{\pi}_{\mathrm{MLP}}$ consisting of a few dense layers, $\tilde{\pi}_{\mathrm{MLP}}$ can then be explained by the obtained salient linguistic terms in a single level manner. We also call $\tilde{\pi}_{\mathrm{MLP}}$ explainable MLP (X-MLP).

**Explainable CNN.** When $\tilde{\pi}$ represents a CNN, say $\tilde{\pi}_{\mathrm{CNN}}$ consisting of convolutional layers with $K$ channels followed by a global average pooling layer. $\tilde{\pi}_{\mathrm{CNN}}$ can then be explained by the obtained salient linguistic terms. Moreover, we also introduce finding out where these salient attributes are related in the given test image $\boldsymbol{X}_j$ by exploiting the spatial information reservation property of the CNN structure. For the $i$-th attribute, $i \in \{1, 2, \ldots, K\}$, its attribute-wise saliency map (i.e., heat map mask) say $\bar{\boldsymbol{X}}_j^i$ can be obtained by the output of the last CNN layer after being upsampled to the same size of $\boldsymbol{X}_j$. In other words, the salient part of $\boldsymbol{X}_j$ corresponding to the $i$-th salient attribute can be shown by $\bar{\boldsymbol{X}}_j^i$.

In contrast to $\tilde{\pi}_{\mathrm{MLP}}$, $\tilde{\pi}_{\mathrm{CNN}}$ provides both the salient linguistic terms and the corresponding attribute-wise saliency maps in a multilevel manner with no extra cost. For ease of reference, we call $\tilde{\pi}_{\mathrm{CNN}}$ explainable CNN (X-CNN).

## 3 EXPERIMENTS

The proposed methodology is trained and tested on coarse-grained and fine-grained benchmark datasets. The data description, implementation setup and results in the XAI regime are given below.

Further details are in the Appendix; in particular, Appendixes B and C are mainly used for further implementation setup and the explainability results, respectively.

**Data.** Animals with Attributes (AwA1) is a well-known dataset used in zero-shot learning (Lampert et al., 2013). Due to the absence of raw images and copyright issues, an alternative version of it named AwA2 was introduced in Xian et al. (2018). It is a medium-scale coarse-grained dataset with 37,322 images from 50 classes collected from public web sources, including 85 attributes per class available. Table 1 presents a size of 5×7 excerpt of AwA2 (see Appendix for the full attribute-class matrix $\mathbf{A}$), exemplifying the nature between the attributes and different classes. The other benchmark dataset used in this work is CUB-200-2011 (Wah et al., 2011). CUB is a fine-grained dataset containing around 11,800 images of 200 different bird classes, including 312 attributes per class available. These linguistic attributes will be exploited to create self-explainable DNNs under our proposed methodology.

Table 1: An excerpt of the attribute-class matrix $\mathbf{A}$ for the AwA2 dataset. Attribute values are in [0, 100] and are standardised before use.

| Attributes / Classes | Gray | Patches | spots | Lean | Tail | Strong | Muscle | $\cdots$ |
|---|---|---|---|---|---|---|---|---|
| Antelope | 12.34 | 16.11 | 9.19 | 39.99 | 40.59 | 33.56 | 26.14 | $\cdots$ |
| Grizzly bear | 3.75 | 1.25 | 0 | 0 | 9.38 | 78.48 | 48.89 | $\cdots$ |
| Killer whale | 1.25 | 68.49 | 32.69 | 22.68 | 41.67 | 63.35 | 10.45 | $\cdots$ |
| Beaver | 7.5 | 0 | 7.5 | 8.75 | 86.56 | 32.81 | 24.38 | $\cdots$ |
| Dalmatian | 0 | 37.08 | 100 | 63.68 | 53.75 | 34.93 | 23.75 | $\cdots$ |

**Implementation setup. I)** For the feature extraction model $\psi$, pre-trained ResNet101 and VGG16 are respectively used for datasets AwA2 and CUB. The sizes of the extracted features for each image in datasets AwA2 and CUB are respectively $8 \times 8 \times 2048$ and $8 \times 8 \times 512$. **II)** The language model $\text{MLP}_\text{L}$ is a few layers wide MLP (here 3 layers are used). To train it, two training sets, $\mathcal{T}$, with size of $5,000$ and $20,000$ respectively for datasets AwA2 and CUB are formed; see Appendix for the details of forming $\mathcal{T}$ in Algorithm 1 and its accuracy performance in Table 3 (on dataset AwA2 where different number of attributes was changed randomly per class). $\text{MLP}_\text{L}$, including the order of the attributes, is frozen after the training completes. **III)** $\tilde{\pi}_{\text{MLP}}$ is a few layers wide MLP (here 4 layers are used) taking 2048 and 512 features extracted by $\psi$ and outputting 85 and 312 attributes for datasets AwA2 and CUB, respectively. $\tilde{\pi}_{\text{CNN}}$ for simplicity is set to one single convolutional layer with the size of $8 \times 8 \times 85$ and $8 \times 8 \times 312$ for datasets AwA2 and CUB, respectively. A $30/70$ split of the data was formed for training/test. **IV)** For comparison, *fine-tuned* ResNet101 and VGG16 are obtained by directly using the training set of image/label pairs of AwA2 and CUB, respectively.

The Adam optimizer with a learning rate of 0.001 and batch size of 32 is used in all experiments. The number of epochs is set to 100 and early stopping is applied (with patience set to 10 based on the validation loss). We stress that the main goal of this work is to make DNNs self-explainable rather than accuracy-driven. It is expected that the prediction performance reported could be improved e.g. with hyper-parameter fine-tuning and/or wiser selection of feature extraction model $\psi$.

**Performance in accuracy.** Table 2 shows that the proposed architecture in Figure 2 can achieve surprisingly good performance (i.e., over $90\%$ and $\sim 50\%$ accuracy for the 50-class and 200-class datasets AwA2 and CUB, respectively) in classification accuracy against the fine-tuned neural networks (i.e., ResNet101 and VGG16 trained directly on the labelled data, which lack explainability) on hold-out test set even though this is not the main aim of this work. The neural networks' perfor-

Table 2: **Classification accuracy**. X-MLP/X-CNN can achieve comparable performance against the fine-tuned ResNet101 and VGG16, which, however, lack linguistic and visual explainability that X-MLP/X-CNN delivers.

| Data | Model | Test Accuracy | Explainability |
|---|---|---|---|
| AwA2 | ResNet101 | $95.8 \pm 1.32$ | N/A |
| | X-MLP | $90.5 \pm 0.84$ | Unilevel |
| | X-CNN | $90.1 \pm 1.12$ | Multilevel |
| CUB | VGG16 | $57.2 \pm 1.44$ | N/A |
| | X-MLP | $54.9 \pm 1.52$ | Unilevel |
| | X-CNN | $44.6 \pm 1.08$ | Multilevel |

mance in accuracy highly depends on the quality of the data acquired. However, most of the data researchers work on, if not all, could be biased, insufficient and/or sensitive. Creating architectures

that can explain themselves and simultaneously reach high prediction performance – just like the one introduced in this work – is arguably the long-term pursuit in machine learning.

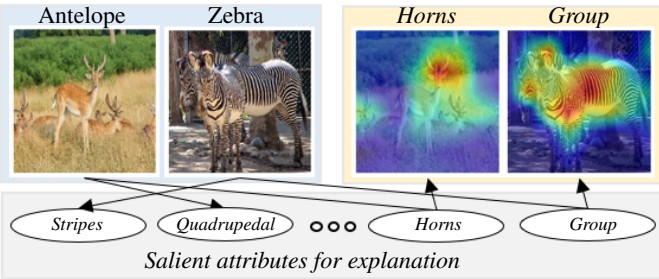

Figure 3: Explainability of the proposed approach for correct prediction. Human-like multilevel explanations are delivered by the salient attributes (by X-MLP/X-CNN) and their saliency maps (by X-CNN), which are matched well.

## 3.1  PERFORMANCE IN EXPLAINABILITY

**Explainablility for correct prediction.** For a given image Least Auklet from the CUB dataset, see Figure 1, both the fine-tuned DNN (VGG16 in our case) and our proposed method can easily classify it correctly. However, the fine-tuned DNN gives no explanation on why it reaches a decision by itself. Post-hoc XAI methods (such as Ribeiro et al. 2016; Selvaraju et al. 2017; Ribeiro et al. 2018) could be employed to see whether the classified object as a whole in the given image is the main part that the fine-tuned DNN focuses on (i.e., left of Figure 1), but this level of explanation is rather limited and is an incomplete reflection of human-like explanations as discussed throughout the article. In contrast, the attribute-wise level of explanation the proposed multilevel XAI model delivers (i.e., right of Figure 1) is much richer, wider, deeper and self-explainable thanks to the linguistic attributes. In detail, some of the most salient attributes that affect the prediction are presented as striped belly, rufous bill and grey breast. Their corresponding saliency maps convincingly highlight the correct part of the image for the mentioned individual attributes. This type of explanation is desirable and is an important indicator of the match between image features and class-wise attributes that are learnt in an unsupervised way by the proposed architecture (unsupervised in the sense that the training images have not been labelled by linguistic attributes and/or salient regions have not been given).

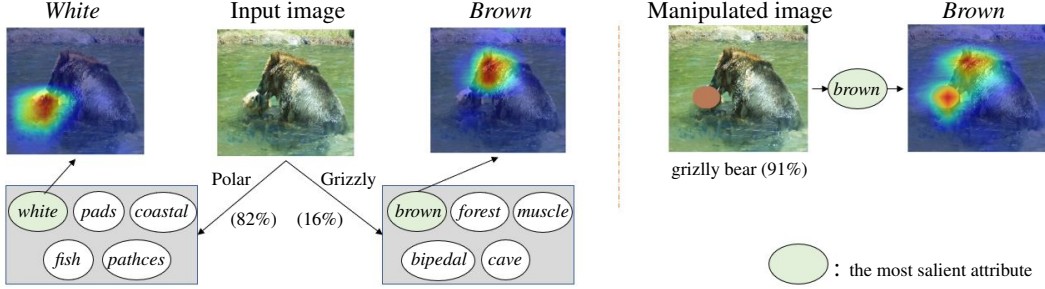

Figure 4: Explainability of the proposed approach for incorrect prediction. Left: A grizzly bear which is misclassified as polar bear. Top five salient attributes are shown in ellipses for the highest probable class (i.e. polar bear) and second class (i.e. grizzly bear). The most salient attributes (i.e., white and brown) and their saliency maps provide insights of the prediction. Right: A manipulated grizzly bear image (obtained by replacing the area related to the attribute "white" by a brown patch) which is then correctly classified by our approach with high confidence (91%).

Figure 3 demonstrates the power of the proposed model in explainability with more challenging images. Linguistic self-explainable attributes of stripes and group are outputted as salient for zebra, while quadrupedal and horns are outputted for antelope by X-MLP and X-CNN. Attribute-wise saliency maps for horn and group outputted by X-CNN show the human-like explanation power of our approach. Further results are provided in Appendix.

**Explainablility for incorrect prediction.** Reaching 100% prediction accuracy is not the case for any method given a nontrivial task. Therefore, investigating the reason behind wrong predictions is

equally important. The left of Figure 4 shows a grizzly bear which is misclassified as polar bear and, to understand the reason behind, the top five salient attributes obtained by the proposed approach for both grizzly bear and polar bear classes are presented. These attributes are indeed the ones that differentiate these two classes (*cf.* the full attribute-class matrix in Appendix). The attribute-wise saliency maps of the most salient attributes (i.e., white and brown) for both classes provide further insights regarding why this misclassification occurred. After occluding the part considered as "white" (by our approach) with a brown patch, the manipulated image is then correctly classified by our approach as grizzly bear with a high confidence, see the right of Figure 4; furthermore, the saliency map for the most salient attribute, "brown", now indeed highlights both the head of the bear and the brown patch, showing that our approach clearly learns what brown is and considers it as a strong indicator of grizzly bear class. Further results are provided in Appendix.

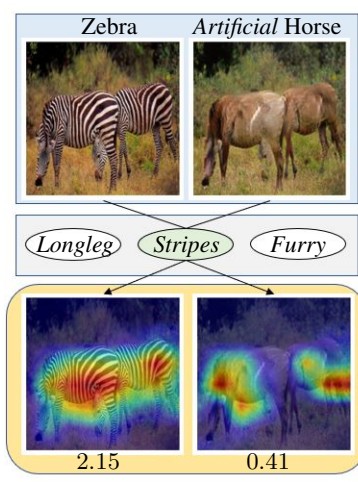

Figure 5: Effectiveness of linguistic attributes in our approach. "Stripes" is one of the salient attributes for zebra with value of 2.15 and its saliency map reasonably highlights the body of zebra. For the artificial horse image, the attribute "stripes" is of value 0.41 and its saliency map is meaningfully unrelated.

**Sensitivity between attributes and features.** To further investigate the effectiveness of the linguistic attributes in our method in explainability, we test a zebra image and its artificial conversion to a horse using CycleGan (Zhu et al., 2017), i.e., the attribute *stripes* is removed from the zebra, see Figure 5. Again, all three models (i.e., fine-tuned ResNet101, X-MLP and X-CNN) classified the zebra image as zebra and the artificially generated horse as horse. At this point the fine-tuned ResNet101 has no explanation ability to show what changed in the original image that forces it to output "horse". In contrast, our model clearly shows that "stripes" is one of the salient attributes for the original zebra image with the attribute value of 2.15 and it drops to 0.41 for the artificially generated horse image. To validate the reason behind this visually, the attribute-wise saliency maps, generated by our approach with *no extra cost*, indicate that the X-CNN model focuses on the body of zebra where the "stripes" lie, whereas arbitrary parts of the artificially generated horse image are highlighted when asked to show where the stripes are, see the bottom of Figure 5.

**Class embeddings with shuffled attribute values.** Previous experiments are conducted using the prior information (e.g., the attribute-class matrix shown in Table 1) provided by experts. An interesting and natural question is: what the results will be if the attribute-class matrix takes different values? In other words, to what extent, will the prior information in the attribute-class matrix be helpful in interpretability? To investigate this, we firstly shuffled all the columns of the attribute-class matrices for both datasets. In an extreme case, say the values of "ground" and "water" for the tiger class may be switched, which apparently would cause a dramatic information loss against the one provided by experts. The shuffled datasets are then used to train the model $MLP_L$. Surprisingly, we found that it converged as fast as using the original data; moreover, the newly trained models X-MLP and X-CNN also reached an accuracy close to the ones obtained by using the original data. Figure 6 shows the interpretability results provided by our approach in this attributes shuffling scenario. Obviously the linguistic attributes become meaningless; moreover, we also observe that the salient regions no longer appear to be associated with the object being recognised and defy an easy human explanation in contrast to what was observed in Figure 1. This finding by our approach shows that those pre-determined attribute lists are crucial to explainability. It also suggests that purely relying on the accuracy of DNNs (which might be trained on data with unknown flaws) could be perilous and the corresponding interpretability is essential. Further discussion is in Section 4.

## 4    DISCUSSION AND LIMITATIONS

To the best of our knowledge, the proposed approach is totally novel to XAI. It also raises a number of questions that are not commonly addressed in this context. Many of these open research questions we believe are important to the further development of XAI. Some of these are outlined below.

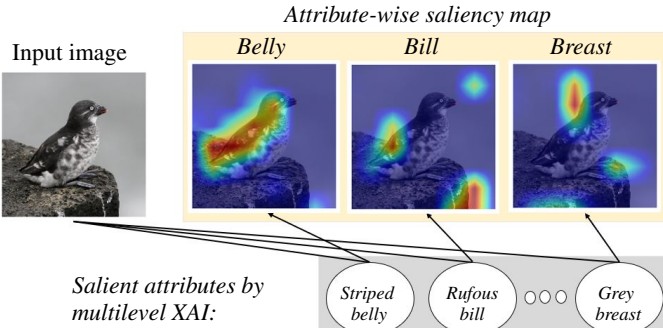

Figure 6: Explainability of the proposed approach in the scenario of attributes shuffling (*cf.* Figure 1). Model accuracy reaches to the one using the true linguistic attribute values, whereas the attribute-wise saliency maps are meaningless to humans, illustrating the importance of prior knowledge of the attribute values.

**Linguistic alignment.** By using a linguistic prior of what classes are associated to a linguistic attribute, we encourage the network to find visual features that correlate with the linguistic attribute – something we call "linguistic alignment". Because we are learning this correlation in an unsupervised manner, we are not guaranteed that the visual feature aligns correctly with the linguistic label. This is likely to improve if we were to use more classes. On a small database of animals dominated by undulates and fish, it would be easy to confuse the linguistic term "hooves" with "legs". Such confusion is much less likely if the dataset contained other animals, such as dogs or primates.

The linguistic alignment is complicated as a DNN is likely to assess circumstantial or contextual evidence for the existence of an attribute. As hooves are highly correlated with legs, it would not be surprising if the legs were considered highly salient to the presence of hooves in an image. This appears to be common in many of the attributes, where saliency maps appear to take in a much larger area than the feature described by the linguistic attribute. At some level, this clearly makes sense. A hoof-like object that is not attached to a leg is unlikely to be a hoof. Similarly, where a hoof is occluded, there may be enough context to infer that the animal is very likely to be hooved. However, this contextual evidence reduces the linguistic alignment. This is a feature of explainability rather than a fault of our approach. However, an important line of research in XAI is to separate circumstantial and contextual evidence from direct evidence.

**The nature of explainability.** Explanations for classifications are not unique. This was shown when we randomised the elements in the matrix, **A**, between classes and linguistic attributes. In doing so, it seems highly unlikely that any attribute has a simple linguistic description. Yet, we can train our model with these random attributes and still obtain classification levels of around $90\%$. This seems at first sight counter-intuitive, although given that we have 85 continuous features they have the potential to carry sufficient information to separate the classes with high accuracy. What separates, at least, some of the true linguistic attributes from other attributes is the information content of the linguistic attributes; that is the linguistic attributes that have a high mutual information in regard to the classes. However, some of these attributes are hard for a DNN to learn.

**What attributes DNNs learn.** We have shown examples of attributes such as "rufous bill" that appear to be well captured by the networks. However, through using an independent test set we find that some of the linguistic attributes appear not to have been learned by the network. An example of this is "big", clearly an attribute that would be useful for humans to distinguish an elephant from a squirrel. Because of the nature of the training set, where objects tend to be resized to fill most of the image, this turns out to carry little information about the classes (see Appendix for more details). However, the lack of success of these attributes is very informative regarding how DNNs perform a discriminative task. All the linguistic attributes used in the data are chosen by experts because linguistically they carry considerable mutual information about the classes. The failure of DNNs to exploit some of these terms obtained by our approach conveys important insights that directly address the issue of what information a neural network is actually learning – a core concern of XAI.

**Tangible *vs* abstract attributes.** The linguistic attributes used in this paper (that we inherited from the zero-shot learning community) interestingly incorporate both tangible and abstract attributes. For the tangible attributes such as "horn" we would expect the corresponding saliency map to highlight horn (although as we have argued it may highlight areas that are important contextual clues to the presence of a horn). The more abstract attributes such as "domestic" or "fast" are less easily

attributed to a particular area of the image. They may however be highly informative for example in differentiating between cat and lion. When these attributes are informative then it is clearly important to understand whereabouts in the image these attribute are inferred. Again our approach goes some way towards addressing this issue.

**Atomic and compound attributes.** In our approach, we have treated all attributes as atomic. Consequently, pointy, fluffy and large ears would all be treated as separate linguistic attributes. However, each attribute would correspond to the same area of the image, and in many cases it seems more natural to treat attributes as compound entities. We have not attempted to do this, but if we wish to scale up our approach to larger datasets, this seems to us to be an important area of future research.

## 5    RELATED WORK AND CONCLUSION

The complexity of machine learning models generally affects the transparency/explainability because of the difficulty of following the model prediction process (Adadi & Berrada, 2018). One line of research is where researchers employ inherently explainable models and utilise white-box models such as bayesian rules (Letham et al., 2015) and linear models (Ustun & Rudin, 2016) to handle complex problems. These models, generally, struggle to reach the prediction ability of DNNs.

Methodologies in the XAI field mainly aim to propose methods to understand how high performance black-box machine/deep learning models work. The majority of XAI methods introduced ideas to explain pre-trained models in a post-hoc manner, i.e., they are neither interested in the training setting nor in changing any of the models' components. These methods could be model-independent requiring only prediction function (Ribeiro et al., 2016; 2018) or model-dependent that need additional information of the trained model such as feature maps at a certain layer (Zhou et al., 2016) or gradients (Selvaraju et al., 2017). DNNs for visual tasks do not output any textual justification. Modern visual-language models are effective in describing image content but lack outputting discriminative features that cause the prediction (Goebel et al., 2018). Forcing these models to output more discriminative features is one related work proposed in Park et al. (2018). It aims to output multilevel explanations for vision-language tasks, e.g., visual question answering and activity recognition. Apart from a completely different focus against the work in this article, this method also requires labour intensive per-image annotations during training that are avoided in our work.

The zero-shot learning regime is where side information (e.g. attributes and class taxonomies) is exploited to classify images of classes that have no labelled samples during training (Xian et al., 2018). The aim is to match image features with class attributes and then to classify unseen classes thanks to the prior side information. There are various of techniques to find the best match that allows unseen class predictions (Frome et al., 2013; Palatucci et al., 2009; Akata et al., 2015; Romera-Paredes & Torr, 2015). Although we integrate side information into our training process similar to zero-shot learning, we are not interested in unseen classes; instead, the proposed work aims to train self-explainable models in many-shot case. Unlike the majority of XAI methods, our explanations are multilevel outputting both linguistic and visual explanations. Finally, different from other extremely limited number of multilevel attempts that specifically work on vision-language models, our training setting is significantly cheaper and does not require per-image annotations.

Concept bottleneck models (Koh et al., 2020) share ideas with our work and are analogous to X-MLP, but neither of them is multilevel. In particular, unlike the concept bottleneck models, our X-MLP and X-CNN only require class-wise attributes, ensuring our approach is significantly cheaper to implement. Moreover, our main focus is the X-CNN, which possesses huge advantages (e.g. see Figure 4) to distinguish our work from the concept bottleneck models.

High performance DNNs are highly desirable when they can reason about their decisions. We presented a new XAI methodology with self-explainable models delivering human-like multilevel explanations alongside the class probabilities. Explaining why a certain prediction is made using linguistic terms and attribute-wise saliency maps without requiring per-image ground-truth explanations in the training phase makes the proposed technique efficient and inexpensive. The results in explainability demonstrated by the match between image features and class embeddings greatly empower the explainability of DNNs while preserving their prediction ability at a reasonable level. Given the importance of XAI and the power of the newly introduced approach, we believe this could spark new avenues in XAI and shed light on developing and applying AI in more sensible ways.

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
