# OpenReview forum: "MULTILEVEL XAI: VISUAL AND LINGUISTIC BONDED EXPLANATIONS"
_ICLR.cc/2023/Conference — Submitted to ICLR 2023_

### Official Review · Reviewer_4txy · 2022-10-23

**Confidence:** 3
**Correctness:** 3
**Technical Novelty And Significance:** 2
**Empirical Novelty And Significance:** Not applicable
**Recommendation:** 5

**Clarity, Quality, Novelty And Reproducibility:**

The paper is well written, original and can be reproduced by the details mentioned in the paper.


**Strength And Weaknesses:**

The paper looks at the important problem of explanation in deep neural networks which are of inevitable importance. Their motivation of defining what is learned by a neural network by breaking down the learning process from coarse categories to fine grained attributes is interesting, intuitive and well supported experimentally.

Despite the interesting approach, there is a limitation as the results and the method are only applied for easy datasets / classes and will be very hard to scale for imagenet classes for instance. What will be the approach to scale for instance the attributes and how would the method perform for thousands of dog breeds in imagenet for example ?

What is the loss function that is used to match the feature and attributes obtained from zero shot literature ?

It is not very clear what the difference is between the single level (Explainable MLP) versus the multilevel Explainable CNN. How’s explaining using attributes (whether with or without saliency maps) a multilevel approach ?

Is there any way to measure the attribute prediction accuracy ?


**Summary Of The Paper:**

The authors propose to learn explanations for the predictions from a neural network by employing a multilevel explanation approach. More specifically, the authors propose to leverage attributes for specific images and map the coarse class labels to the fine-grained object attributes during training. They show that this leads to human-like explanations while achieving nearly similar prediction accuracy. The perform experiments on benchmark datasets and show improved explanation ability of the model compared to state of the art.

**Summary Of The Review:**

The paper is well motivated but the contributions for explainability are marginal and does not bring any significant improvements to the already existing research or outcomes in the field of explainability.

---

> ### Author Response · Authors · 2022-11-10
> **Response to Reviewer 4txy**
>
> Thank you for your review.
>
> Extending the work for large-scale datasets: First of all, it is worth highlighting that our method is not applied to easy datasets only. We would like to remind the reviewer that our network is trained not only on a coarse-grained dataset (i.e., AwA2) but also on the fine-grained CUB dataset which includes 200 bird classes and is quite a challenging dataset. Collecting attributes may seem a drawback and is what we present as the disadvantage of our methodology. However, it is also acknowledged that we can obtain attributes relatively cheaply using pre-trained language models e.g., GPT-3 (or a book on dog breeds). We do not think the Imagenet dog breed subset would be a more challenging task than CUB birds. The number of dog classes in the Imagenet dataset is around 120 out of 1000, which is actually smaller than the bird classes (i.e., 200 classes) in the CUB dataset.
>
> Loss function: The features-to-attributes mapping happens in an unsupervised way thanks to the in-between model X-CNN (i.e.,˜π). Our target is the class label Y; therefore, the loss function is categorical cross-entropy. If we were to learn attributes directly in a supervised way, then we would use binary cross entropy per attribute.
>
> X-MLP X-CNN difference: The main difference is that X-CNN generates both linguistic and visual explanations while X-MLP is limited and creates only linguistic explanations. We trained and presented both results to show that there is freedom of choice, but the resulting explanations would be in different levels. X-CNN adjusts the number of feature maps at the last layer of the model to be equal to the number of linguistic attributes followed by a GAP layer. When we backpropagate a class score with respect to any attributes, we know that each attribute has its feature maps equivalent (e.g., the size of 7×7 or 8×8). This process makes it quite easy to generate saliency maps for attributes. For instance, if the horn is found to be important for the antelope class, we can easily interpolate the feature map to match the size of the input image and highlight what part of the image activates the horn attribute. We can indeed follow a similar process for the X-MLP but it would be much more complicated to backpropagate towards the last CNN layer and make sense of them. For instance, there are 2048 feature maps in the last CNN layer of the model we trained for the AwA2 dataset.
>
> Attribute prediction accuracy: As we only have class-wise attributes, we cannot measure the exact prediction accuracy for attributes. We can measure the prediction accuracy by taking class-wise values as ground truth, but it may be a misleading measurement. The reason is that, for instance, our class-wise attribute map says that elephants have a high value for the trunk attribute but our test image may show an elephant from behind where the trunk is not visible. Let us assume our method predicts that there is no trunk which is the correct prediction, the ground truth would not agree as it is class-wise and the elephant class has got trunk as an attribute. A complete prediction accuracy at the attribute level can only be measured for datasets with attributes per image; otherwise, we can only calculate an approximation using the class-wise matrix.
>
> However, one of our experiments aims to show how much each attribute helps to reduce the uncertainty in the final class, and this experiment can be seen as a way to measure the effect of each attribute (see the mutual information experiment in the Appendix). Moreover, this experiment sheds light on the differences between the way humans and DNNs perceive attributes. For instance, big is found to greatly reduce the uncertainty in the final class for humans (i.e., mutual information calculated with the class-wise ground truth provided by humans); however, when the predicted attribute values are used, the information gain by big/small drops dramatically.
>
> Summary Of The Review: We chose datasets that we felt illustrated the method well, but perhaps they do not emphasise the utility of obtaining saliency maps aligned to human understandable attributes. A better example (although we admit that we have not done this) would be a medical imaging application where we learnt attributes associated with benign cysts and malign tumours. Here we would have the potential to provide a prediction say cancer together with an explanation of a malignant tumour at a particular location, or of being healthy based on only finding a benign cyst at a particular location. We believe that this would be very beneficial clinically. To the best of our knowledge, our approach is the only method that currently provides this kind of information. Certainly compared to one of the mostly widely used XAI models, GradCAM, our model provides information that we believe can lead to far more useful explanations. For this reason, we believe that our method, though simple, is a significant contribution to the field.

---

### Official Review · Reviewer_iU9V · 2022-10-25

**Confidence:** 4
**Correctness:** 3
**Technical Novelty And Significance:** 2
**Empirical Novelty And Significance:** 2
**Recommendation:** 3

**Clarity, Quality, Novelty And Reproducibility:**

**Clarity:** To me, the clarity is poor and feels deliberately obfuscated, especially in the methods section.

**Quality:** See strengths and weaknesses above.

**Novelty:** There is a limited novelty in both the techniques used and the empirical evidence (and discussion).

**Reproducibility:** The method is built on existing models and a code is also released in an anonymized link, I think reproducibility is excellent.

**Strength And Weaknesses:**

**Strengths:**

*[S1:]* Interesting results: The results obtained are certainly interesting; The per-attribute visualization can indeed shed greater light on the decisions of the neural networks compared to "class-level" heat maps.


**Weaknesses:**

*[W1]:* Inadequate baselines, comparisons, and discussion: The proposed visualizations are interesting, but a huge swath of work in vision and language that already demonstrate very similar ability, or can be trivially extended to produce similar visualization and/or linguistic explainability (E.g., concept bottleneck models https://proceedings.mlr.press/v119/koh20a.html). This is woefully missing in the paper and it is very hard to contextualize why the proposed method is more explainable than any number of other works. I will list a few ways this could be done, but the main issue is that it already feels trivial to do what the paper does with existing tools. What new benefits does the proposed work bring?

W1.1. Visualization of existing V&L models: Almost all of the recently introduced VL pretraining methods can ground image regions for a given word in a caption. (E>g., https://github.com/salesforce/ALBEF). Some methods even work for new objects as long as they are adequately described with attributes (A llama, which is a white tall animal in the picture (e.g., https://github.com/microsoft/GLIP)). Going back, even more, most of the object-based representations are also trained to predict object attributes (https://github.com/peteanderson80/bottom-up-attention). The paper makes a distinction between "class-based attributes" and "image-based attributes" but I see no distinction when the attributes are describing various objects; For a "dog" isn't the "class-based attributes" simply a union of all attributes contained for "dog"? (E.g., https://vawdataset.com/). Almost all methods that work with objects and attributes can visualize their affinity in various image regions to some degree. How does this compare with the proposed method?

W1.2 Trivial modification of existing work (E.g., Grad-CAM) to use additional information used by the paper: The paper makes use of additional attributes to obtain the given visualizations; However, if any network was modified to predict these attributes in addition to the class label, Grad-CAM visualization for making the prediction "has beak", could be used in exactly the same way the paper is using.

Without a comparison, discussion, or adequate comparative analysis with these, the impacts of the paper are hard to justify.

*[W2]:* The effect of components used is not explained: Firstly, the method requires additional annotations. While the paper does mention that may be easy to obtain (e.g., via large language models such as GPT-3), there are no experiments showing whether a lower-quality (noisy) attribute annotations would still work. Similarly, the training of MLP_L is not described well. How sensitive is it? What happens when only a small portion of the attributes are present? What happens when attributes (y_hat) is not predictive of class y? Since having high-quality attribute data seem to be the central requirement


*[W3]:* Lack of Clarity: The paper feels obfuscated and very hard to follow. However, upon closer inspection, the core concepts are fairly simple. I would urge the authors to streamline the presentation and explain in terms of their function rather than hiding details unnecessarily under hard-to-follow conventions. For example, in Section 2.2 paragraph 2, the first sentence is literally just describing the process to generate predictions from a neural network. It is entirely unnecessary to write that as:

> 8Xj 2 X used for test, let yXj = †(˜⇡( (Xj ))), i.e., the predicted class label of the test image Xj. (Not rendered properly here -- please see pdf.)

The second part of the sentence beginning with i.e. describes the same thing as the stuff before that. This is simply an example in what is an unnecessarily dense symbol used to describe fairly simple underlying processes in Section 2 (Methodology).


**Summary Of The Paper:**

The paper proposes a new explainable neural network architecture called Multilevel XAI architecture. The key contribution of the work is to use both visual and language-guided interpretability, where a list of attributes pertaining to a class prediction (e.g., Dog --is furry, ...). Experiments show the model's ability to ground salient attributes in images in addition to providing region-based saliency maps.

**Summary Of The Review:**

I am leaning toward recommending the rejection of this paper. While there certainly are interesting tidbits, the biggest gripe is that the proposed model requires both class label and attribute annotation per class; If we allow other methods to have access to the same (or similar) type of data, they could (and do) produce similar "interpretable" visualizations as shown in the paper.

---

> ### Author Response · Authors · 2022-11-10
> **Response to Reviewer iU9V**
>
> Thank you for your review.
>
> [W1]: Inadequate baselines, comparisons, and discussion: We would like to thank the reviewer for making us aware of the “concept bottleneck models”. Clearly, this work shares ideas with our own and we will, of course, cite it in our paper. However, the novelty and main contributions of our methodology remain the same, since concept bottleneck models are analogous to X-MLP in our work, neither of them are multilevel. However, our X-MLP only requires class-wise attributes, unlike the concept bottleneck models. Thus our approach is also far less expensive to implement. Moreover, our main focus, as clear in the title, is the X-CNN (linguistic and visual) which proposes many advantages to distinguish our work from the concept bottleneck models.
>
> The concept bottleneck models are able to output unilevel concepts as explanations whereas our technique outputs more detailed explanations, i.e., linguistic and visual. Although it may seem trivial to extend the results in the referenced paper to generate attribute-wise maps as we do, the authors did not attempt to do that. There may have been several reasons why they did not and our discussion section sheds light on the potential issues one can face when attempting to visualise the learnt concepts. Simply ignoring the visual equivalent of the given concept and claiming that, for example, this bird is a Herring Gull because it has got black under tail colour is incomplete and maybe even misleading (see Figure 3 in the concept bottleneck models paper). How can one be sure that the model actually learns what black under tail colour is? This highlights the main strength of our paper: we presented plenty of examples to show that our model learns both classes through concepts and concepts themselves without requiring a separate concept learning model. We even transparently showed what concepts are not learnt in the way we expected (see the Appendix). Moreover, we discussed potential reasons why saliency maps for some attributes may be different from what we expected in the Discussion section.
>
> The main finding in the concept bottleneck models paper is that you can intervene on the model and change the predicted concept values in the concept space to see the change in the final prediction. For instance, Figure 7 in their paper shows that if you reduce the predicted value of white throat and increase the one of the yellow throat, the final prediction will go from wrong to correct. How can one be sure white throat or yellow throat is actually corresponding to the throat region? We believe that the similar experiment we presented in Figure 4 in our paper provides considerably more information than the main finding of the concept bottleneck models. In Figure 4, our model tells us that the reason why a brown bear is predicted as a white bear is due to the white colour in the image and points out where the white region is. We then change the concept (e.g., white) in the input space, not in the concept space, and the prediction turns out to be correct. We convincingly showed that changing the given concept (e.g., adding a brown patch on the white region) will correct the final prediction. Since we do this manipulation in the input space, what part we change in the input image is clear. Whereas it is still a black-box approach to claim that tweaking the predicted throat value (just like the concept bottleneck models do) changes the prediction, we simply do not know if they are actually changing the signal coming from the throat region or anything else that may be correlated with it. We think that our multilevel explanations are much more meaningful and make it easy to intervene on the model predictions in the input space.
>
> In addition, concept bottleneck models require considerably more annotations than our method does, i.e., instance-level attribute annotations. In practice, this makes concept bottleneck models much harder to train and implement.
>
> Please see the follow-up comment for our response to the rest of your review.

---

> ### Author Response · Authors · 2022-11-10
> **Response to Reviewer iU9V (Second part)**
>
> W1.1. Visualization of existing VL models: We do not think it is fair and relevant to compare our XAI methodology with VL models. We present our work for image classification task to compete with well-known methodologies that dominate the XAI field, such as GradCAM. Currently, GradCAM is used widely and people try to make sense of generated class-wise maps while our method handles this issue by creating attribute-wise maps.
>
> One reason why we cannot compare our model with VL models is because ours is significantly cheaper than VL models as it does not require any instance-level attribute annotation, segmentation, bounding box or text prompts. Some of these annotations are a must in different VL models. For instance, CLIP (https://openai.com/blog/clip/) requires images paired with text and takes an enormous amount of computation and data to train. We do not think VL models are efficient to be used for explaining image classification results. VL models may reach similar results but much more expensively. We would like to hear if the reviewer knows any VL method that only requires class-wise attribute annotations in addition to the raw images similar to our approach.
>
> There is a clear distinction between class-wise and image-wise attributes. We want to emphasize that we do not obtain class-wise attributes by unifying/ averaging the per-image attributes because we do not have access to per-image annotations. For instance, for the AwA2 dataset, we only have access to the class-wise attributes; e.g., antelopes have horns, they are wild, etc. Obtaining image-wise attributes is expensive, whereas class-wise attributes can easily be obtained from various linguistic sources, e.g., by simply asking Google what an antelope looks like and using the given answers as attributes.
>
> W1.2 Trivial modification of existing work (E.g., Grad-CAM) to use additional information used by the paper: We do not claim that we propose a complicated methodology. We tried to keep our architecture and training setting as simple as possible. Modifying networks to introduce an attribute layer for interpretable models is what we propose. Our model generates multilevel explanations quite cheaply and its results are impressive compared to the well-known methodologies which only create class-wise maps.
>
> [W2]: The effect of components used is not explained: This is one of our future works. We are currently working on scenarios where we do not have high-quality attributes or we are missing a part of attributes for some classes. However, the attributes are not high quality even for the datasets we used. For example, the chimpanzee class has a high value for the tail attribute which is wrong. In addition, the experiments on the mutual information in the Appendix show a lot of attributes that provide almost 0 information in reducing the uncertainty. Therefore, we would like to emphasise that the attribute-class matrices that we use are already noisy.
>
> W3]: Lack of Clarity: We understand that our method section may seem dense and can be simplified. We are happy to convert technical details and make them easier to read. However, we did not deliberately obfuscate anything. We tried to formalise the details to avoid any vagueness or ambiguity. We wanted readers not to be confused and see exactly what is proposed as the architecture and how each component functions. We also explained what we meant by the given mathematical equations.
>
> Novelty: We are confused and disappointed. This is the first time, to the best of our knowledge, an XAI technique links attributes to saliency maps in XAI settings. In our opinion, this substantially improves the explainability power of our method over existing methods. We kindly ask the reviewer to look at Figure 5 where a zebra image is artificially converted to a horse. While the methodologies in the field such as GradCAM would only highlight the body of the animal giving no insight into what has changed; our methodology clearly says that the value of stripes has dropped as well as showing where the stripe signal comes from. We chose a somewhat artificial domain, which perhaps does not sufficiently illustrate why this is so important. However, suppose we were working in medical imaging, being able to distinguish a benign cyst from a malign tumour has hugely more explanatory value than giving saliency maps attached to unlabelled feature maps. Compared to the concept bottleneck model our method would be able to highlight the part of the image where it predicted there was a cyst or tumour. Thus we believe that, although simple, our contribution is quite significant for explainable AI systems.

---

### Official Review · Reviewer_Snsk · 2022-10-27

**Confidence:** 4
**Correctness:** 2
**Technical Novelty And Significance:** 3
**Empirical Novelty And Significance:** 2
**Recommendation:** 3

**Clarity, Quality, Novelty And Reproducibility:**

The idea is interesting. Writing is easy to follow. Reproducibility is good due to the provided code.

**Strength And Weaknesses:**

**Strength**

1 Using multilevel (visual and linguistic) to conduct more intuitive explanations is an interesting and promising idea.

2 The author shows an easy-to-follow method and results.

**Weaknesses**

1 Generalization. The proposed method could only be used to explain a specific kind of NN, which learn a mapping starting from an extracted feature to an attribute-related label embedding. It is hard to explain general pre-trained neural networks (e.g., ResNet, GNN, Transformers).

2 Partial explainable. The whole model has three parts, feature extractor, added learnable mapping, and attribute to label mapping; only the middle part can be explained, and the best part still performs as a black box.

3 Fidelity issue. The method treats not the true label Y but an attribute set as a target during training. How to guarantee the GPT-3 described attributes are useful features for prediction? If you directly train a model using true label Y, they use different features during the decision. So the explained attributes do friendly to human understanding but may not follow the original model's logic.

4 Need to improve accuracy results. The final accuracy is lower and has a relatively large gap compared with the original model, even though the proposed method uses more learnable parameters. These results also aligned with point 3 above. The newly learned explainable mapping does not use the same logic as the original model.

5 Accumulated bias. There may be some bias in the feature extractor model (Resnet) and GPT-3. The new trained mapping can not remove those biases if they are input and target.

**Summary Of The Paper:**

This paper proposes a visual and linguistic bounded explanation method to make part of NN models explainable by adding both attribute-wise and language-wise explanations. Specifically, they add a trainable part between the feature extractor (e.g., pre-trained Resnet) and label embedding, and the added part can be explained. The paper is easy to follow and clear.

**Summary Of The Review:**

Overall, using a multilevel explanation is interesting, while the reviewer thinks the weakness outweighs the strength.

---

> ### Author Response · Authors · 2022-11-10
> **Response to Reviewer Snsk**
>
> Thank you for your review.
>
> 1 Generalization: We aim to train self-explainable DNNs and our proposed method is an initial attempt, working well for image classification tasks. Any kind of extensions can be seen as future work. We invite researchers to use our methodology and come up with ideas on how it can be applied to GNNs, Transformers or any other kind of NN. Please note that we used ResNet and VGG16 in our experiments. In principle, we see no obstacle to using our technique for visual transformers where we introduce an attribute layer and investigate what segments of an image are important in activating a particular attribute. We have not come across any XAI methodology that works for any given DNNs. Different techniques in the XAI field are generally proposed to work on one or a few tasks, following that others expand them further for even more challenging settings.
>
> 2 Partial Explainable: Our approach provides an explanation of why a particular decision is made in terms of human-understandable attributes such as horn, stripe, black, etc. It also provides the saliency maps associated with these attributes. We believe that this adds a level of explainability that other widely used XAI methods such as GradCAM lack. There are questions about the full workings of a network that our approach does not attempt to capture. We are sure you are aware there are methods such as visualising the inputs that excite internal layers of a network that have been widely explored. This is an interesting work that provides insights into how deep learning networks work; however, this is not what we are trying to do and we would argue it is somewhat orthogonal to the great majority of XAI approaches. Possibly we are misunderstanding what you mean as this seems a criticism that would apply to almost all XAI techniques. If so we apologise and would welcome additional explanation of why the reviewer believes our approach suffers from partial explainability in a way that other approaches do not.
>
> 3 Fidelity issue and 4 Need to improve accuracy results: We would like to clarify that we do not train an auxiliary model that aims to explain the predictions of a fine-tuned model (we could do that, but it is not how we are thinking this approach will be used). Rather we are developing a machine that both provides predictions and explanations. Therefore, there is no fidelity issue. Table 2 only aims to compare the performance results, and the original fine-tuned model is not included in any of the experiments for explanation generation. Moreover, we presented many examples for different attributes in the Appendix to show that our explanations are not only valid for a single prediction or handpicked. We accept that there is a gap between the performance of our model and that of the original model. We felt that gap to be rather small and did not put any effort into hyper-parameter tuning or attribute selection as we wanted to present the raw model in its simplest form. We acknowledge that the attributes we use may fail to capture aspects that contributed to excellent performance of the original network. This is an area that we intend to explore further in future work, although we do not see this as a game-changer (the performance we obtain is well beyond that easily achievable without using deep learning). For clarity, we reiterate that we first train the MLP L that maps the attributes to the classes and freeze its parameters. After that, the X-CNN takes high-level image features as inputs and image classes Y as labels. During training, the X-MLP or X-CNN map image features to attributes without having the attribute labels.
>
> Please see the next comment for our response to Accumulated bias and Rating. We have to submit two separate comments due to the character limit.

---

> ### Author Response · Authors · 2022-11-10
> **Response to Reviewer Snsk (Second part)**
>
> 5 Accumulated bias: It is possible that biases in the dataset could be compounded by biases in the attribute labelling, but our technique provides a quite powerful tool for detecting biases. For instance, we realised that the chimpanzee class has a high value for the tail attribute in the provided attribute-class matrix which is wrong. Moreover, the mutual information experiment presented in the Appendix showed the differences between the way humans and DNNs perceive the attributes. The big/small size-related attributes are found to have a negligible mutual information value which is initially surprising (although understandable when we realise that the images are usually scaled so the object fills the frame). Therefore, we think that our method indeed helps to find out and reduce the bias in the data.
>
> Rating: We kindly ask the reviewer to clarify why they think several of the paper’s claims are incorrect or not well-supported and point out what these claims are so that we can improve our work. It is clear that the reviewer rates our work with a quite low mark due to concerns that are open research questions in XAI and not the claims of our paper such as partial explanations, bias and fidelity issues. We wrote the paper from the perspective of developing a method that would help users of deep-learning systems to understand the classification being made by providing human-understandable attributes with saliency maps corresponding to those attributes. This would be useful in say a medical imaging application where we would like to know if a shadow on an X-ray was interpreted as a malignant tumour or a benign cyst. To the best of our knowledge, this is the only XAI approach that provides these two levels of information. It seems that the view of the reviewer is that XAI is there to understand the full workings of a deep networks. This is undoubtedly an important problem and we believe that our approach can contribute to a small part of this understanding (e.g. we can investigate how important direct evidence from the object is in making a classification compared to circumstantial evidence from contextual information), but this was not the intention of the current paper. We hope that our response will be helpful, and we are looking forward to further discussion.

---

### Decision · Program_Chairs · 2023-01-20

**Decision:**

Reject

**Justification For Why Not Higher Score:**

The paper follows an interesting direction. However, all the concerns raised by the reviewers suggest that the paper is not mature enough to be published.

**Justification For Why Not Lower Score:**

N/A

**Metareview: Summary, Strengths And Weaknesses:**

This paper presents an explainability approach for CNNs. The approach combines class activation maps with the semantic representation learned by the model to produce explanations based on salient attributes and attribute-wise salient maps. The experiments are conducted with the AWA2 and CUB-200-2011 datasets.

The reviewers value the interest of producing explanations that combine visual and semantic information, and the insights observed in the obtained results. However, the reviewers also pointed out the following aspects: (1) The approach is not general enough to be applied to any pre-trained NN; (2) Need to test the proposed approach in larger scale datasets; (3) Novelty with respect to existing methods is limited. The work also requires a better contextualization of the proposed approach with respect to existing methods and more comparisons with related methods. Despite the feedback of the authors the main concerns still remain, making the paper to be not mature enough to be published at ICLR. We encourage the authors to extend the related work adding the references recommended by the reviewers. We also recommend the authors to extend their experiments to larger scale datasets and even to the medical imaging domain, as suggested by the authors in their feedback.